# In vivo reprogramming drives *Kras*-induced cancer development

Hirofumi Shibata[1,2], Shingo Komura[1], Yosuke Yamada[1,3], Nao Sankoda[4], Akito Tanaka[1], Tomoyo Ukai[1], Mio Kabata[1], Satoko Sakurai[1], Bunya Kuze[2], Knut Woltjen [1,5], Hironori Haga[3], Yatsuji Ito[2], Yoshiya Kawaguchi[4], Takuya Yamamoto [1,6] & Yasuhiro Yamada[1,6,7]

The faithful shutdown of the somatic program occurs in the early stage of reprogramming. Here, we examined the effect of in vivo reprogramming on *Kras*-induced cancer development. We show that the transient expression of reprogramming factors (1–3 days) in pancreatic acinar cells results in the transient repression of acinar cell enhancers, which are similarly observed in pancreatitis. We next demonstrate that *Kras* and *p53* mutations are insufficient to induce ERK signaling in the pancreas. Notably, the transient expression of reprogramming factors in *Kras* mutant mice is sufficient to induce the robust and persistent activation of ERK signaling in acinar cells and rapid formation of pancreatic ductal adenocarcinoma. In contrast, the forced expression of acinar cell-related transcription factors inhibits the pancreatitis-induced activation of ERK signaling and development of precancerous lesions in *Kras*-mutated acinar cells. These results underscore a crucial role of dedifferentiation-associated epigenetic regulations in the initiation of pancreatic cancers.

[1] Department of Life Science Frontiers, Center for iPS Cell Research and Application (CiRA), Kyoto University, Kyoto 606-8507, Japan. [2] Department of Otolaryngology, Gifu University Graduate School of Medicine, Gifu 501-1194, Japan. [3] Department of Diagnostic Pathology, Kyoto University Hospital, Kyoto 606-8507, Japan. [4] Department of Clinical Application, Center for iPS Cell Research and Application (CiRA), Kyoto University, Kyoto 606-8507, Japan. [5] Hakubi Center for Advanced Research, Kyoto University, Kyoto 606-8501, Japan. [6] AMED-CREST, AMED, 1-7-1 Otemachi, Chiyodaku, Tokyo 100-0004, Japan. [7] Division of Stem Cell Pathology, Center for Experimental Medicine and Systems Biology, Institute of Medical Science, University of Tokyo, Tokyo 108-8639, Japan. Correspondence and requests for materials should be addressed to Y.Y. (email: yasu@ims.u-tokyo.ac.jp)

nduced pluripotent stem cells (iPSCs) can be established from somatic cells by the transient expression of pluripotency-related transcription factors (TFs)[1]. Given that epigenetic regulation plays a central role in cell fate determination and maintenance, this technology allows for the reorganization of epigenetic regulation from the differentiated somatic cell state into the pluripotent state without affecting genetic information[2–4]. Recent studies revealed the stepwise epigenetic changes during iPSC derivation[5,6]. In early-stage reprogramming, the faithful shutdown of the somatic program occurs through the silencing of somatic cell-specific enhancers, which is associated with a loss of original cell identity. Upon prolonged expression of the reprogramming factors, gradual activation of the pluripotency-related transcriptional network is observed in late-stage reprogramming.

Notably, such events in cellular reprogramming are similarly observed in cancer development. Cancer cells often lack terminal differentiation ability, suggesting that repression of the somatic cell program is involved in cancer development. In contrast, poorly differentiated cancers that lose most differentiation features sometimes exhibit an activation of pluripotency-related genes such as targets of *Nanog*, *Oct3/4*, and *Sox2* (ref. [7]). Moreover, poorly differentiated cancers with pluripotency-related gene expressions are associated with high malignancy, especially in the mammary glands[7]. Collectively, these findings suggest that some aspects of cellular reprogramming are associated with cancer development.

Indeed, our previous study demonstrated that premature termination of in vivo reprogramming leads to kidney cancer development through altered epigenetic regulation[8]. Consistent with the partial reprogramming state, these cancer cells lose kidney cell-specific molecular signatures while they partially acquire the trait of embryonic stem cells (ESCs) including self-renewing capacity. Notably, these cancers resemble Wilms' tumor, which is the most common childhood kidney cancer. Furthermore, these cancer cells were readily reprogrammable into iPSCs that are capable of differentiating into non-cancerous kidney cells[8]. These results raised the possibility that reprogramming-associated epigenetic regulation has a significant impact on childhood cancer development, which is also in agreement with recent observations that childhood cancers harbor relatively few genetic mutations. However, the functional significance of epigenetic regulation related to cellular reprogramming remains largely unclear in adult cancer development.

Pancreatic cancer is one of the most common causes of cancer mortalities in adults in developed countries. The median survival period is less than 6 months and the 5-year survival rate is 3–5% [9,10]. The most common type of pancreatic cancer is pancreatic ductal adenocarcinoma (PDAC). PDAC is one of the well-characterized cancers for multistep cancer progression models that have "Big 4" mutations (mutations in *KRAS*, *CDKN2A*, *TP53*, and *SMAD4*)[11,12]. Histologically, PDAC is thought to arise from a spectrum of preneoplastic lesions with ductal morphology that are designated as pancreatic intraepithelial neoplasia (PanIN) [13]. Since early PanIN lesions harbor mutations or amplification of the *KRAS* at high frequency (over 90%), it has been proposed that *KRAS* mutation is responsible for PanIN formation and thus is an initial event during pancreatic cancer development[14,15]. In contrast to premalignant lesions, PDAC often harbors additional mutations such as a loss of *CDKN2A* and inactivating mutations at *TP53* and *SMAD4*[11]. Together, current consensus holds that pancreatic cancer arises through a stepwise progression with characteristic genetic mutations for each histologically distinguishable stage.

In the present study, based on the shared properties between cellular reprogramming and cancer development, we examined the effect of in vivo reprogramming in the development of *Kras*-induced cancer. We show that reprogramming-mediated repression of somatic cell enhancers, which is associated with early stage of reprogramming, in conjunction with *Kras* mutation results in rapid development of PDAC. We also show that partial acquisition of the ESC signature, which occurs later stage of reprogramming, causes the development of cancers that resemble human α-fetoprotein (AFP)-producing cancer. These results highlight the crucial role of reprogramming-related epigenetic regulation in *Kras*-induced cancer in vivo.

## Results

**Kras and p53 mutations are insufficient for ERK activation**. We first generated *Pdx1-ires-Cre* knocked-in ESCs by homologous recombination (Fig. 1a and Supplementary Fig. 1A), and then *Pdx1-ires-Cre* knocked-in mice to induce pancreas-specific Cre-*LoxP* recombination. Lineage tracing analysis of *Pdx1-ires-Cre* mouse using *Rosa26 LacZ* reporter allele[16] confirmed that the Cre-*LoxP* recombination occurs in almost all pancreatic cells (Supplementary Fig. 1B, C), which was consistent with a previous report using transgenic mice expressing *Pdx1-Cre*[17]. Similarly, we knocked-in *LoxP-Stop-loxP (LSL)-HA tag-Kras^{G12D}* allele into endogenous *Kras* locus in ESCs (Supplementary Fig. 1D) and generated knocked-in mice. We then generated *LSL-HA tag-Kras^{G12D}* and *Pdx1-ires-Cre* (hereafter *HA-KC*) compound mice to examine the expression of *Kras^{G12D}* oncoprotein (Fig. 1a). HA immunostaining exhibited membranous expression of *Kras^{G12D}* oncoprotein in the pancreatic cells of *HA-KC* mice at 6 weeks of age (Fig. 1b). However, despite the expression of *Kras^{G12D}* oncoprotein, most pancreatic cells were histologically normal (Fig. 1b) except for the focal formation of early PanIN, indicating that *Kras* mutation alone cannot transform pancreatic acinar cells.

To further investigate the effect of oncogenic mutations on the ERK signaling pathway and aberrant proliferation, we next generated *LSL-Kras^{G12D}, Pdx1-ires-Cre* (KC) mice and *LSL-Kras^{G12D}, LSL-p53^{R172H}, Pdx1-ires-Cre* (KPC) compound mice (Fig. 1c) using well-established *Kras* and *p53* mutant alleles[18,19]. In accordance with the results in *HA-KC* mice, most pancreatic cells exhibited recombination in both *Kras^{G12D}* and *p53^{R172H}* alleles in *KPC* mouse (Fig. 1d, Supplementary Fig. 9). However, the majority of pancreatic cells were histologically normal in both KC and KPC mice at 6 weeks of age, except for the spotted formation of early PanIN and the dysplastic area (Fig. 1e). Although PDAC development was observed in *KPC* mice at 8 weeks of age, the affected area was still limited, and most pancreatic cells remained histologically normal (Fig. 1e). Consistently, pERK immunostaining showed positive staining only in PanIN and PDAC lesions, while the majority of pancreatic cells were negative for pERK (Fig. 1e), demonstrating that compound mutations of *Kra*s and *p53* are insufficient for robust activation of the ERK pathway. Similarly, the majority of pancreatic cells in *KPC* mice were negative for Ki67, a cell proliferation marker (Fig. 1e), which indicates that compound mutations of *Kra*s and *p53* gene are insufficient for inducing the abnormal proliferation of pancreatic cells. These results suggest that aberrations beyond *Kras* and *p53* mutations are required for PDAC development.

**Reversible ductal metaplasia by in vivo reprogramming**. In order to investigate the influence of in vivo reprogramming in the pancreas, we next generated mice in which reprogramming factors can be induced specifically in the pancreas upon doxycycline (Dox) treatment. We established mice harboring *Pdx1-ires-Cre, Rosa LSL-rtTA3*, and *Col1a1::tetO-Oct3/4, Sox2, Klf4, c-Myc-ires-mCherry* (hereafter *C-OSKM*) (Fig. 2a). After Dox treatment, we confirmed mCherry fluorescence in the pancreas of these mice

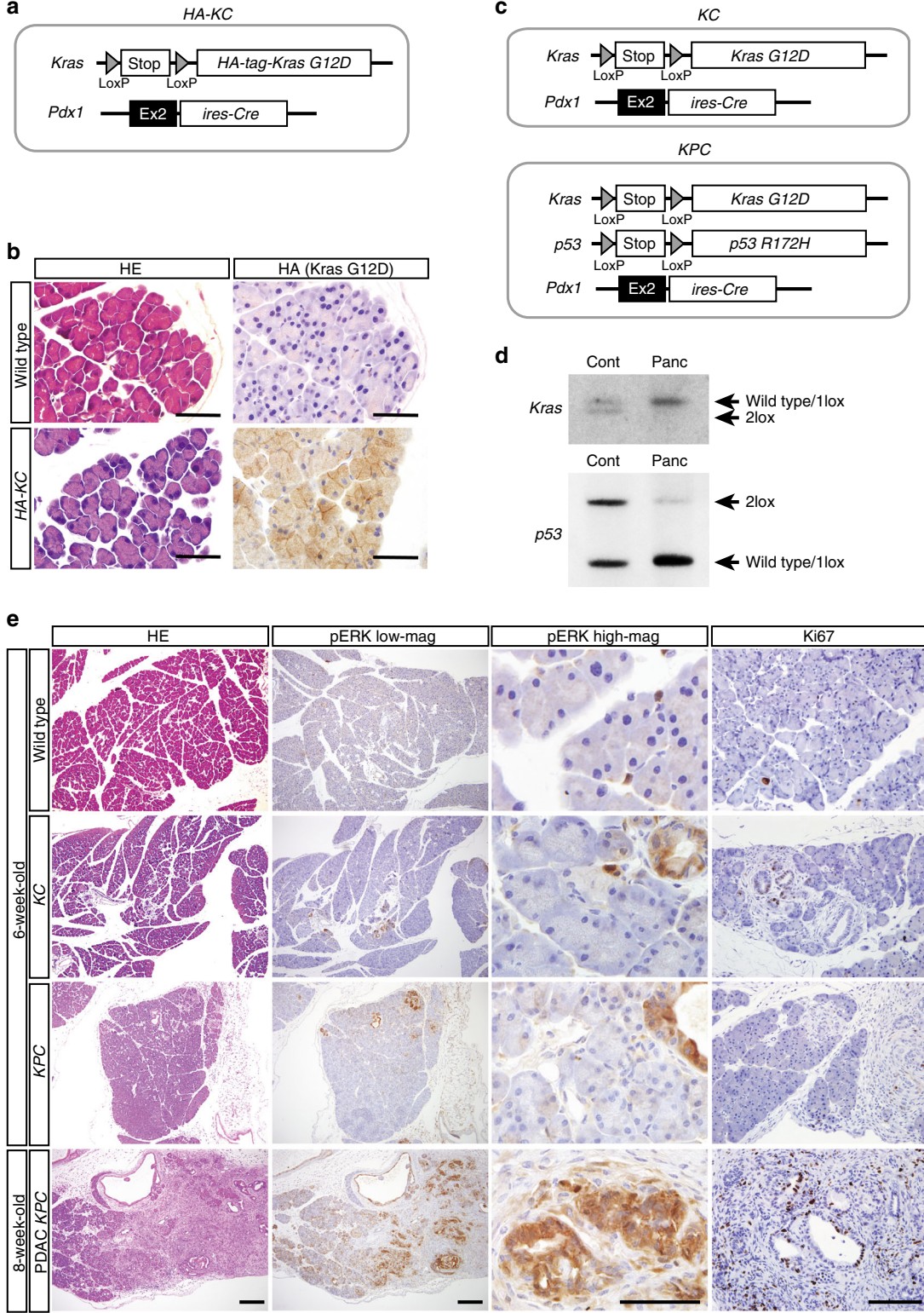

**Fig. 1** *Kras/p53* compound mutations are insufficient for PDAC development. **a** A schematic illustration of the genetic construct to activate *HA-Kras^G12D* in the pancreas. **b** Representative histology and immunostaining for *HA-Kras^G12D*. Scale bars, 50 μm. **c** Genetic strategy to study the influence of somatic activation of *Kras* and *p53* mutations in the pancreas. **d** Southern blotting of *LSL-Kras^G12D* and *LSL-p53^R172H* allele. ESCs containing *Pdx1 ires-Cre*, *LSL-Kras^G12D* and *LSL-p53^R172H* alleles were used as control. Note that majority of two *LoxP* alleles are converted into one *LoxP* allele in the pancreas of *KPC* mouse (Cont control, Panc pancreas). **e** Immunostaining for pERK and Ki67 in the pancreas of 6-week-old wild-type mice, *KC* mice, and *KPC* mice. *KPC* mice at 8 weeks of age showed a focal PDAC area with pERK staining (bottom). Scale bars: HE and pERK (low magnification) staining, 200 μm; pERK (high magnification) staining, 50 μm; and Ki67 staining, 100 μm

(Supplementary Fig. 2A). We also performed histological analysis of 4-week-old mice given Dox for 3 days (Fig. 2b).

It has been proposed that the transition of acinar cells into a duct-like state (acinar to ductal metaplasia (ADM)) is an early event of pancreatic cancer development[20–22]. Notably, Dox-treated mice often exhibited dilatation of the acinar glands and expression of CK19, a ductal marker (Fig. 2c). CK19 expression

was sometimes observed in amylase-expressing cells (Fig. 2c), suggesting that ADM occurs by the short induction of reprogramming factors. Consistent with this notion, Oct3/4 expression was observed in newly formed ductal cells (Fig. 2d). Notably, after 1 week of Dox withdrawal, the histology of the pancreas was almost normal, and CK19-expressing ADM lesions were hardly detected (Fig. 2e, f), presumably reflecting the

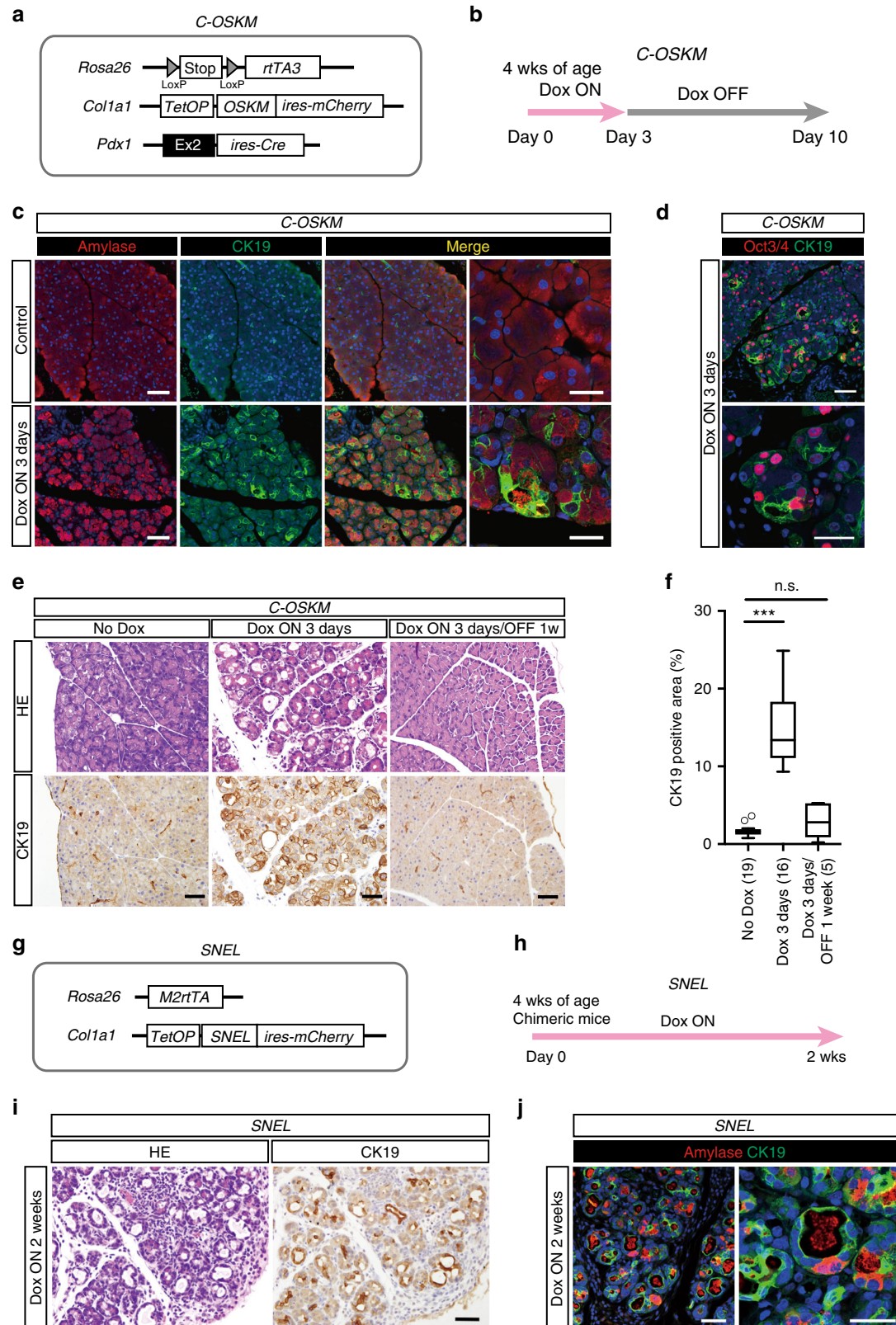

epigenetic memory of acinar cells. Taken together, these results show that the 3-day expression of reprogramming factors causes reversible ADM phenotype.

A recent study demonstrated that continuous *Klf4* over-expression induces ADM[23]. To further investigate whether cellular reprogramming is indeed related to ADM formation, we next examined the effect of another reprogramming cocktail that lacks *Klf4*. For this purpose, we established ESCs with Dox-inducible polycistronic cassette containing *Sall4, Nanog, Esrrb*, and *Lin28* (hereafter *SNEL*), which was reported as an alternative reprogramming factor cocktail, using the KH2-ESC system (Fig. 2g)[24,25]. To investigate the effect of *SNEL* expressions on reprogramming, mouse embryonic fibroblasts (MEFs) containing Dox-inducible *SNEL* alleles were established from E13.5 embryos. After 2 weeks of Dox exposure to *SNEL*-MEFs, iPSC-like colonies were observed (Supplementary Fig. 2B, C). Although these iPSC-like colonies could not be expanded, they exhibited increased alkaline phosphatase activity (Supplementary Fig. 2D), reflecting the partial reprogramming state. We then examined the in vivo effect of *SNEL* expression in the pancreas. After Dox treatment, *SNEL*-inducible chimeric mice exhibited mCherry fluorescence in various organs, including the pancreas (Supplementary Fig. 2E, F). Notably, Dox treatment longer than 2 weeks resulted in the formation of ADM lesions in the pancreas (Fig. 2h, i). Consistently, CK19-positive regions often co-expressed amylase (Fig. 2j), indicating that ADM was similarly induced by *SNEL* expression. Moreover, the ADM lesions were hardly detectable at 1 week after Dox withdrawal, suggesting that the *SNEL*-induced ADM phenotype is reversible (Supplementary Fig. 2G). Collectively, these results indicate that the transient expression of reprogramming factor cocktails induces reversible ADM phenotype.

**In vivo reprogramming represses acinar cell enhancers**. To investigate the molecular basis of reprogramming-mediated ADM formation, we first examined transcriptional changes of the pancreas during in vivo reprogramming. The pancreas of *C-OSKM* mice given Dox for 3 days was analyzed for global gene expressions by microarray. In accordance with histological evidence of ADM formation, the expression of acinar cell-related TFs such as *Ptf1a* and *Mist1* (*Bhlha15*) was reduced, while the expression of *CK19* (*Krt19*), a ductal cell-specific gene, was markedly increased (Fig. 3a). In addition, ADM-related genes such as *Muc5ac, Tff1*, and *Plac8* were similarly upregulated (Fig. 3a)[26,27]. To examine the initial transcriptional response after *OSKM* induction, we next treated *C-OSKM* mice with Dox via i.p. injection and examined gene expressions after 24 h. Notably, marked repression of *Ptf1a* and *Mist1* was already detectable at 24 h after Dox treatment (Fig. 3b). Conversely, *CK19* and ADM-related genes were upregulated in these samples (Fig. 3b),

suggesting that the silencing of acinar cell-related genes is coupled with the induction of ADM-related genes.

Previous studies demonstrated that silencing of the somatic cell-specific enhancers is an initial epigenetic event during reprogramming[5,6]. We next examined the effect of reprogramming factor expressions on the enhancers of pancreatic cells. ChIP-seq analysis for H3K27ac was performed using the pancreas of *C-OSKM* mice given Dox for 3 days. A perfusion fixation protocol was employed for cross-linking chromatin in vivo. Consistent with reduced expressions, H3K27ac deposition was significantly reduced at the enhancers of several acinar cell-related genes including *Ptf1a, Mist1* (*Bhlha15*), *Nr5a2*, and *Cpa1* (Fig. 3c and Supplementary Fig. 3A) after Dox treatment, but it was not altered at the control *Actb* gene (Supplementary Fig. 3B). In contrast, H3K27ac deposition was not increased in upregulated genes, such as *CK19* or in ADM-related genes (Supplementary Fig. 3C), which supports the notion that the repression of acinar cell enhancers is an initial epigenetic regulation step after reprogramming factor expression. Similarly, *Sox9*, a key TF in ductal cells[22,28], did not exhibit an enrichment of H3K27ac deposition (Supplementary Fig. 3C). Furthermore, genome-wide analysis revealed that enhancers including super enhancers (SEs)[29,30] are significantly repressed after Dox treatment for 3 days (Fig. 3d and Supplementary Fig. 3D, E). Consistently, the number of enhancers and SEs were substantially decreased by Dox treatment for 3 days (Fig. 3d and Supplementary Fig. 3E). Moreover, the expressions of genes linked with Dox OFF-specific SEs were decreased by Dox treatment, while genes linked with common-SEs or Dox ON-specific SEs were not repressed (Fig. 3e). *Ptf1a* promoter and enhancer remained unmethylated in the pancreas after 3 days of Dox treatment (Supplementary Fig. 3F), indicating that DNA methylation was not involved in the *Ptf1a* repression. Notably, the Dox-induced reduction of H3K27ac deposition was not observed at 2 weeks after Dox withdrawal, indicating the repression of SEs is reversible (Fig. 3d). Collectively, these results suggest that reprogramming factor expression causes transient epigenetic repression of acinar cell enhancers and loss of acinar cell identity, which lead to ADM formation. To further connect the loss of acinar cell enhancer with ADM formation, we induced acinar-cell-related TFs (*Mist1* or *Ptf1a*) together with *OSKM* in the pancreas (Supplementary Fig. 4A–C). Notably, the co-expression of *Ptf1a* or *Mist1* with *OSKM* attenuated the *OSKM*-induced expression of CK19 and ADM formation (Fig. 3f and Supplementary Fig. 4D). Together, these results suggest that the silencing of acinar cell program causes the induction of ADM.

Finally, to connect *OSKM*-induced ADM formation with pathological conditions, we examined the effect of caerulein treatment, which is well-established pancreatitis stimuli and induces transient ADM formation, on pancreatic SEs. We

**Fig. 2** Transient expression of reprogramming factors induces reversible acinar to ductal metaplasia. **a** A schematic illustration of the genetic construct of Pdx1 ires-Cre, Rosa LSL-rtTA3, Col1a1::tetO-OSKM-ires-mCherry (C-OSKM) mice. **b** Experimental protocol for Dox treatment in *C-OSKM* mice. **c** Immunofluorescent images for Amylase and CK19 in the pancreas of *C-OSKM* mice. CK19 expression is observed in the acinar area of *C-OSKM* mice after Dox treatment for 3 days. Scale bars, 50 μm. **d** Immunofluorescent images for Oct3/4 and CK19 in the pancreas of *C-OSKM* mice. Transgenic Oct3/4 expression is coincident with the CK19-positive lesions. Scale bars, 50 μm. **e** Reversible ductal metaplasia by the transient expression of reprogramming factors. CK19 staining is hardly detected in the acinar area after Dox withdrawal for 1 week. Scale bars, 50 μm. **f** Quantification of the CK19-positive area in the pancreas of *C-OSKM* mice. A box-and-whisker plot of the CK19-positive area. Solid lines in each box indicate the median. Bottom and top of the box are lower and upper quartiles, respectively. Whiskers extend to ±1.5 interquartile range (IQR). Numbers in parentheses indicate the number of mice examined. ***$p < 0.001$, Mann–Whiteney *U*-test. n.s. not significant. **g** A schematic illustration of the genetic constructs of *Rosa M2-rtTA, Col1a1::tetO-SNEL-ires-mCherry* (KH2-SNEL) mice. Polycistronic cassette was used for *SNEL* induction. **h** Experimental protocol for Dox treatment in KH2-*SNEL* chimeric mice. **i** CK19 expression in Dox-treated KH2-*SNEL* chimeric mice. ADM lesions are observed after *SNEL* expression for 2 weeks. Scale bar, 50 μm. **j** Immunofluorescent images for Amylase and CK19 staining in KH2-*SNEL* chimeric mice. CK19 expression is observed in the amylase-positive acinar area after *SNEL* expression. Scale bars, 50 μm

performed intraperitoneal injection of caerulein eight times for 2 days (2 μg/injection every consecutive hour for 7 h) and performed ChIP-seq analysis for H3K27ac after 24 h at Day 3. Histological analysis revealed that *OSKM*-expressing pancreatic lesions phenocopies caerulein-induced ADM lesions

(Supplementary Fig. 4E). Consistent with previous studies, we confirmed that the caerulein treatment causes reduced expression of acinar cell-related genes such as *Ptf1a* and *Mist1* (Supplementary Fig. 4F). Of particular note, enhancers that are altered by *OSKM* expression were similarly altered by caerulein treatment

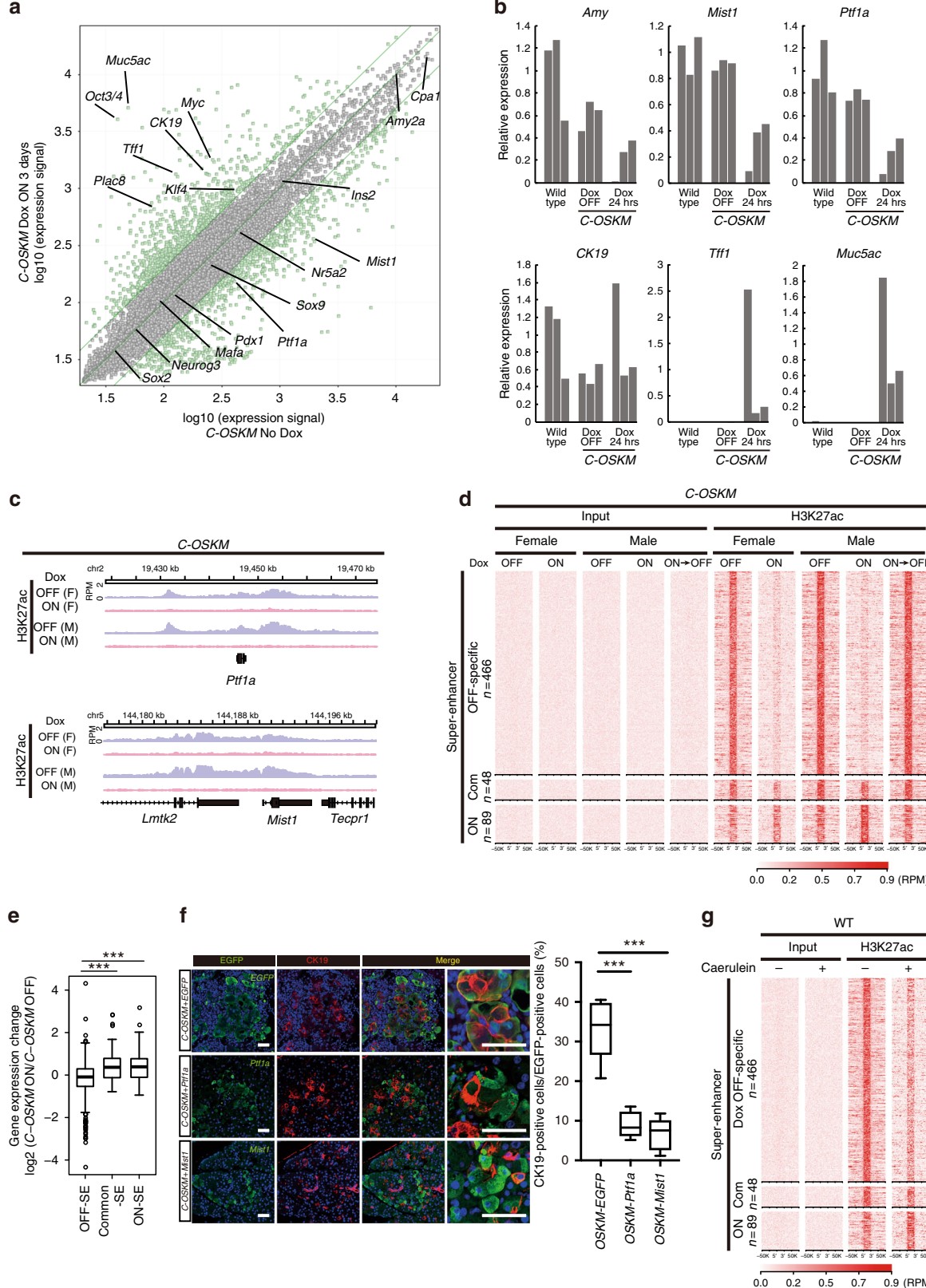

(Fig. 3g and Supplementary Fig. 4G). Taken together, our results suggest that inflammatory stimuli induce ADM formation through repression of somatic enhancers.

**In vivo reprogramming activates ERK in *Kras* mutant pancreas.** To elucidate the influence of the loss of acinar cell identity on *Kras*-mutated pancreatic cells, we next established ESCs containing *LSL-Kras^{G12D}*, *Pdx1-ires-Cre*, *Rosa LSL-rtTA3*, and *Col1a1::tetO-OSKM-ires-mCherry* (hereafter *KC-OSKM*) and generated chimeric mice to induce *Kras* mutation and reprogramming factors simultaneously in the pancreas (Fig. 4a). The experimental protocol of Dox treatment for *OSKM* induction is shown in Fig. 4b. Consistent with our findings that *OSKM* expression induces ADM, increased expression of CK19 was observed in both *C-OSKM* mice and *KC-OSKM* mice (Fig. 4c). Notably, pERK was strongly induced in CK19-positive area of *KC-OSKM* mice whereas such induction was modest in *C-OSKM* mice (Fig. 4c), suggesting that *Kras* mutation cooperatively activates the ERK pathway in conjunction with reprogramming factor expression (Fig. 4c). In sharp contrast to reversible ductal metaplasia in *C-OSKM* mice, CK19-expressing ducts were observed even at 1 week after Dox withdrawal in *KC-OSKM* mice (Fig. 4d), suggesting that the ADM phenotype is sustained in the presence of *Kras* mutation. Consistently, increased expression of pERK was sustained in the ductal area at 1 week after Dox withdrawal in *KC-OSKM* mice (Fig. 4e, f). Furthermore, ductal lesions were histologically displayed PanIN/PDAC (Fig. 4d, e, Supplementary Fig. 5A). The affected lesions sometimes displayed positive staining for Alcian blue, a PanIN marker, and often exhibited widespread Sirius red staining, which showed that massive fibrosis simultaneously occurs in PanIN/PDAC lesions within 10 days of Dox exposure (Fig. 4e, f, Supplementary Fig. 5B).

Previous studies have demonstrated that increased expression of endogenous EGFR activates the MAPK pathway in human and mouse PanIN/PDAC area[31,32]. To unveil the molecular mechanism for persistent activation of the ERK signaling pathway in *KC-OSKM* mice, we examined EGFR expression in Dox-treated *C-OSKM* and *KC-OSKM* mice. Although Dox treatment for 3 days increased EGFR expression in both *C-OSKM* and *KC-OSKM* mice, the increased EGFR levels were pronounced in *KC-OSKM* mice, while only modest expression was observed in *C-OSKM* mice (Fig. 4g and Supplementary Fig. 5C). Furthermore, the increased EGFR expression was sustained in *KC-OSKM* mice even after Dox withdrawal, whereas *C-OSKM* mice showed almost no EGFR staining (Fig. 4g and Supplementary Fig. 5C, 5D), suggesting that *Kras* mutation is involved in robust and continuous EGFR expression through positive feedback regulation. Western blot confirmed the sustained activation of the ERK signaling pathway and increased EGFR expression in the pancreas of *KC-OSKM* mice even after Dox withdrawal (Fig. 4h,

Supplementary Fig. 9). These results suggest that maintenance of the acinar cell trait prevents ERK signaling activation in the pancreas of *Kras* mutant mice.

**p53 mutation accelerates *Kras/OSKM*-induced PDAC.** An inactivating mutation at the *p53* gene was reported to promote PDAC formation[33,34]. To investigate the effect of *p53* mutation on in vivo reprogramming-mediated development of pancreatic lesions in *Kras* mutant background, we next generated chimeric mice containing *LSL-Kras^{G12D}*, *LSL-p53^{R172H}*, *Pdx1-ires-Cre*, *Rosa LSL-rtTA3*, and *Col1a1::tetO-OSKM-ires-mCherry* (hereafter *KPC-OSKM*) (Fig. 5a). The time course of Dox treatment is shown in Fig. 5b. Transient induction of *OSKM* in the pancreas of *KPC-OSKM* mice significantly increased the CK19- and pERK-positive area, which persisted even after withdrawal of Dox (Supplementary Fig. 5E, F) as similarly observed in *KC-OSKM* mice. Notably, *KPC-OSKM* mice developed predominantly PDAC within 10 days of *OSKM* induction, while *KC-OSKM* mice developed mixed lesions containing both PanIN and PDAC (Fig. 5c–e). Consistently, Dox-treated *KPC-OSKM* mice showed a significant reduction of PanIN-related Alcian blue-positive area when compared with Dox-treated *KC-OSKM* mice (Supplementary Fig. 5G). Dox-treated *KPC-OSKM* mice died within 3 months, and some mice developed cancerous ascites (Supplementary Fig. 5H). Moreover, cell lines established from pancreatic tumors in Dox-treated *KPC-OSKM* mice were able to develop secondary tumors in the subcutaneous tissue of nude mice (Supplementary Fig. 5I), affirming that they have cancer-initiating properties. Together, these results demonstrate that compound mutations of *Kras* and *p53* in cooperation with transient repression of acinar cell enhancers are sufficient for the rapid and widespread induction of PDAC.

We further examined mechanistic insights into the phenotypic difference between *KC-OSKM* and *KPC-OSKM* mice after Dox exposure. Given that escape from *Kras*-induced senescence is associated with progression of PanIN and that the *p21–p53* pathway is involved in oncogene-induced senescence[35], we particularly focused on the effect of *p53^{R172H}* mutation on cellular senescence. Consistent with the previous report that found *OSKM* expression in vivo induces cellular senescence[36], widespread DNA damage response, which was measured by γH2AX staining, as well as increased *p21* expression were detected after Dox treatment in *C-OSKM* mice (Supplementary Fig. 5J). Consistently, an increased deposition of H3K27ac was observed in *Cdkn1a* (*p21*) after Dox treatment for 3 days in *C-OSKM* mice (Supplementary Fig. 5K). Notably, the induction of γH2AX and *p21* was indistinguishable among *C-OSKM*, *KC-OSKM*, and *KPC-OSKM* mice (Supplementary Fig. 5J), suggesting that oncogenic *Kras* mutation and *p53* mutation hardly affect the immediate senescence-associated response against reprogramming factor expressions. After withdrawal of

**Fig. 3** Repression of acinar cell enhancers by in vivo reprogramming. **a** Microarray analysis in the pancreas of *C-OSKM* mice given Dox for 3 days. **b** A qRT-PCR for acinar cell-related genes and ADM-related genes after Dox treatment for 24 h in the pancreas of *C-OSKM* mice. Data are presented as the mean of technical triplicates. The mean expression level of wild-type samples was set to 1 for *Amy*, *Mist1*, *Ptf1a*, and *CK19*. For *Tff1* and *Muc5ac*, the mean expression level of Dox 24 h samples was set to 1. **c** ChIP-seq analysis for H3K27ac in the pancreas of *C-OSKM* mice. Note that H3K27ac deposition is substantially reduced at acinar cell-related transcription factors (*Ptf1a* and *Mist1*). **d** Super enhancers (SEs) in the pancreas of *C-OSKM* mice. Note that Dox OFF-specific SEs show significant reduction in H3K27ac deposition in Dox ON pancreas. The number of SEs is decreased after Dox treatment for 3 days. **e** Dox OFF-specific SE-related gene expressions are significantly suppressed by *OSKM* induction. ***p < 0.001, Mann–Whiteney *U*-test. **f** Forced expression of an acinar cell-related transcription factor (*Ptf1a* or *Mist1*) inhibits *OSKM*-induced ADM formation. EGFP expression represents the expression of either *Ptf1a* or *Mist1*. Note that most EGFP-expressing cells are negative for CK19 while the control EGFP-expressing cells often costained with CK19. Scale bars, 50 μm. The ratio of CK19-positive cells in EGFP-positive cells of 10 independent images are shown in a box-and-whisker plot. Solid lines in each box indicate the median. Bottom and top of the box are lower and upper quartiles, respectively. Whiskers extend to ±1.5 interquartile range (IQR). ***p < 0.001, Mann–Whiteney *U*-test. **g** Super enhancers (SEs) in caerulein-treated pancreas. Note that caerulein treatment causes reduction in H3K27ac deposition at SEs that are repressed by *OSKM* induction

Dox for 1 week, however, the *p21-* and γH2AX-positive area was substantially reduced in proliferating lesions of both *KC-OSKM* and *KPC-OSKM* mice (Supplementary Fig. 6A–C), suggesting that escape from senescence occurs regardless of the *p53* status. Consistently, there was no significant difference in the survival rate between Dox-treated *KC-OSKM* mice and Dox-treated

*KPC-OSKM* mice (Supplementary Fig. 6D), suggesting that *Kras* mutation alone together with transient repression of acinar cell enhancers cause the escape from senescence and the rapid PDAC development.

Taken together, our results demonstrate that the repression of acinar cell enhancers results in ADM formation and the

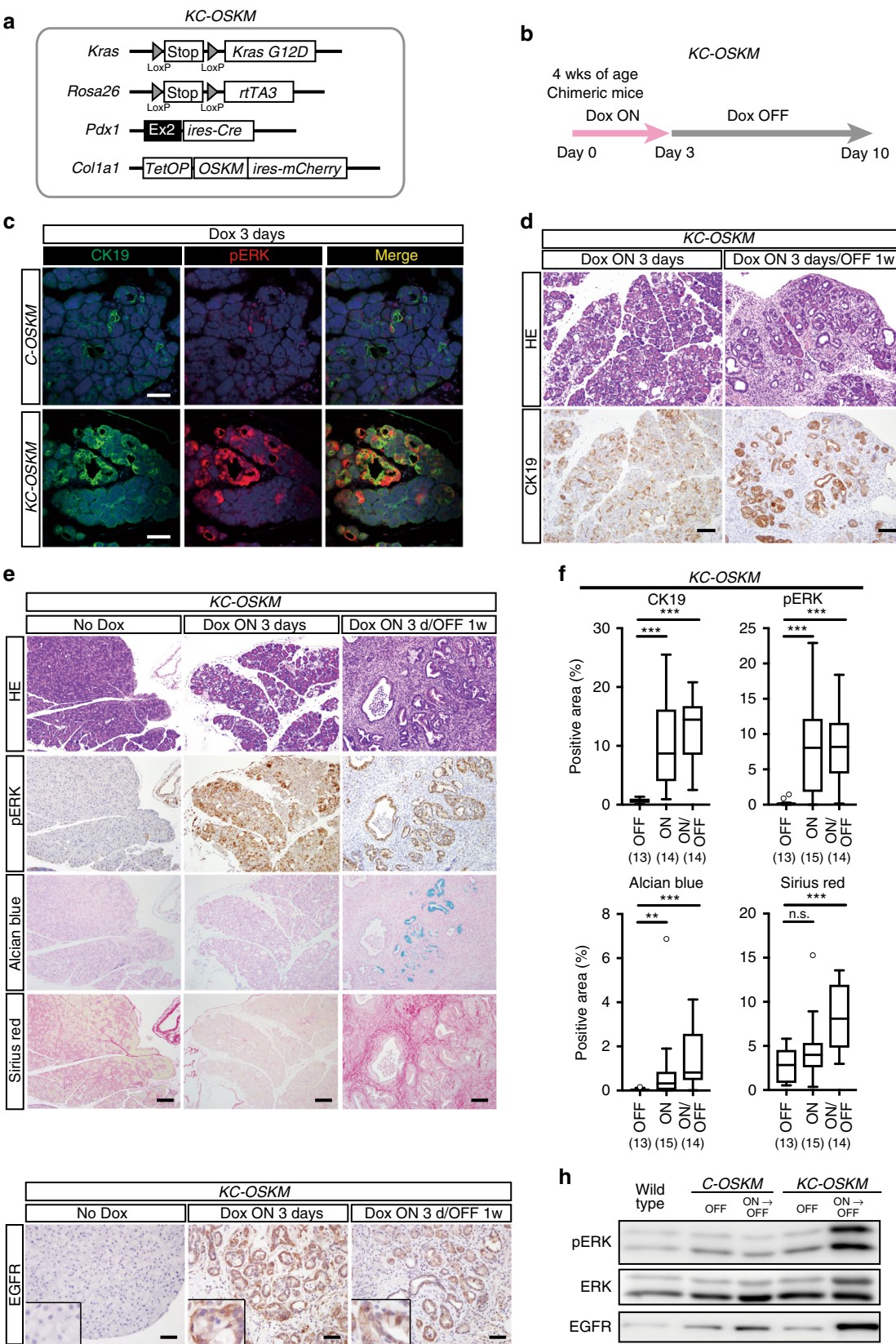

subsequent cancer growth in *Kras*-mutated acinar cells of the pancreas. Finally, we tried to maintain acinar cell enhancers by expressing acinar cell-specific TFs (*Ptf1a* or *Mist1*) in caerulein-treated *KC* mice, in which caerulein-induced inflammation initiates ADM/PanIN formation and activates ERK signaling (Fig. 5f and Supplementary Fig. 4E). Notably, forced expression of either *Ptf1a* or *Mist1*, but not the control *mCherry*, strongly inhibited PanIN formation (Fig. 5g and Supplementary Fig. 7A). Consistently, pERK staining was significantly reduced by the expression of acinar cell-related TFs (Fig. 5h). These results provide additional evidence that acinar cell enhancers safeguard *Kras*-induced cancer initiation.

**Undifferentiated cancer development by in vivo reprogramming.**
It has been shown that the acquisition of the pluripotency-related epigenetic landscape occurs at late-stage reprogramming[5,6]. We next extended the duration of reprogramming factor expression in *KC-OSKM* mice to examine the effect of late epigenetic regulation of reprogramming on cancer development. *KC-OSKM* mice were given Dox for 1–2 weeks, followed by Dox withdrawal for 1–2 weeks. In contrast to the PanIN/PDAC formation in *KC-OSKM* mice given Dox for 3 days, prolonged treatment of Dox caused the development of histologically undifferentiated cancer with higher proliferative activity in the pancreas of *KC-OSKM* mice (Fig. 6a). Notably, these cancers exhibited the expression of pluripotency-related proteins, Sall4 and Lin28a, independently of *OSKM* transgene expressions (Fig. 6a).

We found that *KC-OSKM* mice given Dox for 1–2 weeks often developed undifferentiated cancers in the stomach (Fig. 6b and Supplementary Table 1), which is consistent with the observation that *Cre-LoxP* recombination also occurs in the pyloric glands of the stomach with *Pdx1-ires-Cre* allele (Supplementary Fig. 1B, C). These gastric cancers similarly exhibited the expression of Sall4 and Lin28a (Fig. 6c). In agreement with the acquisition of pluripotency-related epigenetic regulation, partial activation of ESC-Core and ESC-Myc modules was observed in gastric cancers in *KC-OSKM* mice (Supplementary Fig. 8A). *C-OSKM* mice treated with Dox for 1–2 weeks also developed undifferentiated cancer in the stomach (Supplementary Fig. 8B, C), albeit at lower frequency than *KC-OSKM* mice ($p < 0.01$, Fisher's exact test, Supplementary Table 1), suggesting that the *Kras* mutation promotes but is not essential for the development of undifferentiated cancers.

**In vivo reprogramming induces AFP-producing cancers.**
Human AFP-producing cancers predominantly arise in the stomach and display poorly differentiated histology and show dismal prognosis[37,38]. Notably, we observed focal expression of Afp in gastric cancers in *KC-OSKM* mice given Dox for 1–2 weeks (Fig. 6c, d), which raised the possibility that the acquisition of pluripotency-related epigenetic regulation is associated with the

development of human AFP-producing cancers. Therefore, we next investigated the expression of pluripotency-related proteins in human AFP-producing gastric cancers. Twenty-five gastric cancers that were diagnosed as AFP-producing gastric cancers, were examined for the expression of SALL4, LIN28A, and LIN28B by immunostaining. Notably, as observed in gastric cancers in *KC-OSKM* mice, SALL4, LIN28A, and LIN28B were frequently co-expressed in human AFP-producing gastric cancers (Fig. 6e, f and Supplementary Table 2), suggesting that these cancers exhibit activation of the pluripotency-related regulatory network.

Finally, we re-examined the microarray data previously used to investigate 200 cases of human gastric cancers[39]. We found that *AFP*, *LIN28B*, and *SALL4* were overexpressed in seven, eight, and nine cases of the 200 human gastric cancers, respectively. Consistent with the frequent co-expression of pluripotency-related proteins in human AFP-producing gastric cancers, partial activation of ESC-Core and ESC-Myc modules was observed in *AFP*-, *LIN28B*-, and *SALL4*-expressing human gastric cancers (Fig. 6g and Supplementary Fig. 8D), while such activations were not obvious in *ERBB2*-highly expressed gastric cancers ($n = 12$) (Supplementary Fig. 8D), which recapitulated our observations of reprogramming-induced gastric cancers in *KC-OSKM* mice.

## Discussion
Cancer is believed to arise through an accumulation of genetic mutations[40,41]. Indeed, genome-wide sequencing analyses have revealed a number of driver mutations in multiple cancer types. The functional significance of such driver mutations has been demonstrated by reverse genetic approaches where rodents in which these mutations were introduced develop the same types of cancer observed in human patients. However, the genetically engineered cancer models always exhibit a substantial latency period for cancer development, indicating that additional events are required for the development of full-blown cancer. Therefore, which sets of genetic mutations are sufficient for cancer development remains unclear in most cancers. Indeed, in the present study we show that most pancreatic cells with compound mutations at *Kras* and *p53* display no obvious activation of the ERK signaling pathway and exhibit normal histology at 6 weeks of age, which is consistent with previous findings that oncogenic *KRAS* mutations are detectable in human disease-free pancreas[42]. Notably, we demonstrate that short, transient expression of reprogramming factors in *Kras* mutant mice at 4 weeks of age results in persistent ERK activation and widespread PDAC development within 10 days, indicating that cellular reprogramming is sufficient for the initiation of cancer growth in pancreatic cells with *Kras* mutation.

Alteration of the epigenetic modifications is one of the most common aberrations detected in human cancers[43–45]. Decades of studies have demonstrated that epigenetic regulation plays a significant role in cancer development both in vitro and in vivo.

**Fig. 4** Transient expression of *OSKM* causes persistent activation of ERK signaling and PDAC/PanIN development in the *Kras*-mutated pancreas. **a** A schematic illustration of the genetic construct of *Pdx1 ires-Cre, LSL-Kras^{G12D}, Rosa LSL-rtTA3, Col1a1::tetO-OSKM-ires-mCherry* (*KC-OSKM*) mice to induce pancreas-specific *Kras* mutation and in vivo reprogramming. **b** Experimental protocol for Dox treatment in *KC-OSKM* chimeric mice. **c** Immunofluorescent images for CK19 and pERK staining. Note that strong pERK staining is detectable in ADM lesions of *KC-OSKM* mice. Scale bars, 50 μm. **d** CK19 immunostaining of the pancreas in *KC-OSKM* chimeric mice. CK19-expressing cells are observed even after Dox withdrawal for 1 week. Scale bars, 100 μm. **e** Representative images for HE, pERK, Alcian blue, and Sirius red staining in the pancreas of *KC-OSKM* chimeric mice. Scale bars, 100 μm. **f** Quantification of CK19-, pERK-, Alcian blue-, and Sirius red-positive areas in the pancreas of *KC-OSKM* chimeric mice. A box-and-whisker plot of the positive area. Solid lines in each box indicate the median. Bottom and top of the box are lower and upper quartiles, respectively. Whiskers extend to ±1.5 interquartile range (IQR). Numbers in parentheses indicate the number of mice examined. **p < 0.01, ***p < 0.001, Mann–Whiteney *U*-test. n.s. not significant. **g** EGFR immunostaining in the pancreas of *KC-OSKM* chimeric mice. Note that increased EGFR expression is induced by Dox treatment and sustained after Dox withdrawal. Insets represent a higher magnification. Scale bars, 50 μm. **h** Western blot analysis for pERK, ERK, and EGFR of the pancreas. Increased and sustained pERK and EGFR expression are observed in *KC-OSKM* mice

However, recent sequencing analysis identified that most cancer types harbor mutations at epigenetic modifier genes at varying frequencies, which has raised the possibility that genetic abnormalities are the primary cause of the dysfunction or dys-regulation of epigenetic modifiers that result in secondary alterations in the cancer epigenome. Therefore, direct evidence of bona fide epigenetic alterations that drive cancer development is still limited, especially in vivo.

iPSC derivation is mediated through epigenetic reorganization without affecting genomic information. In the initial stage of

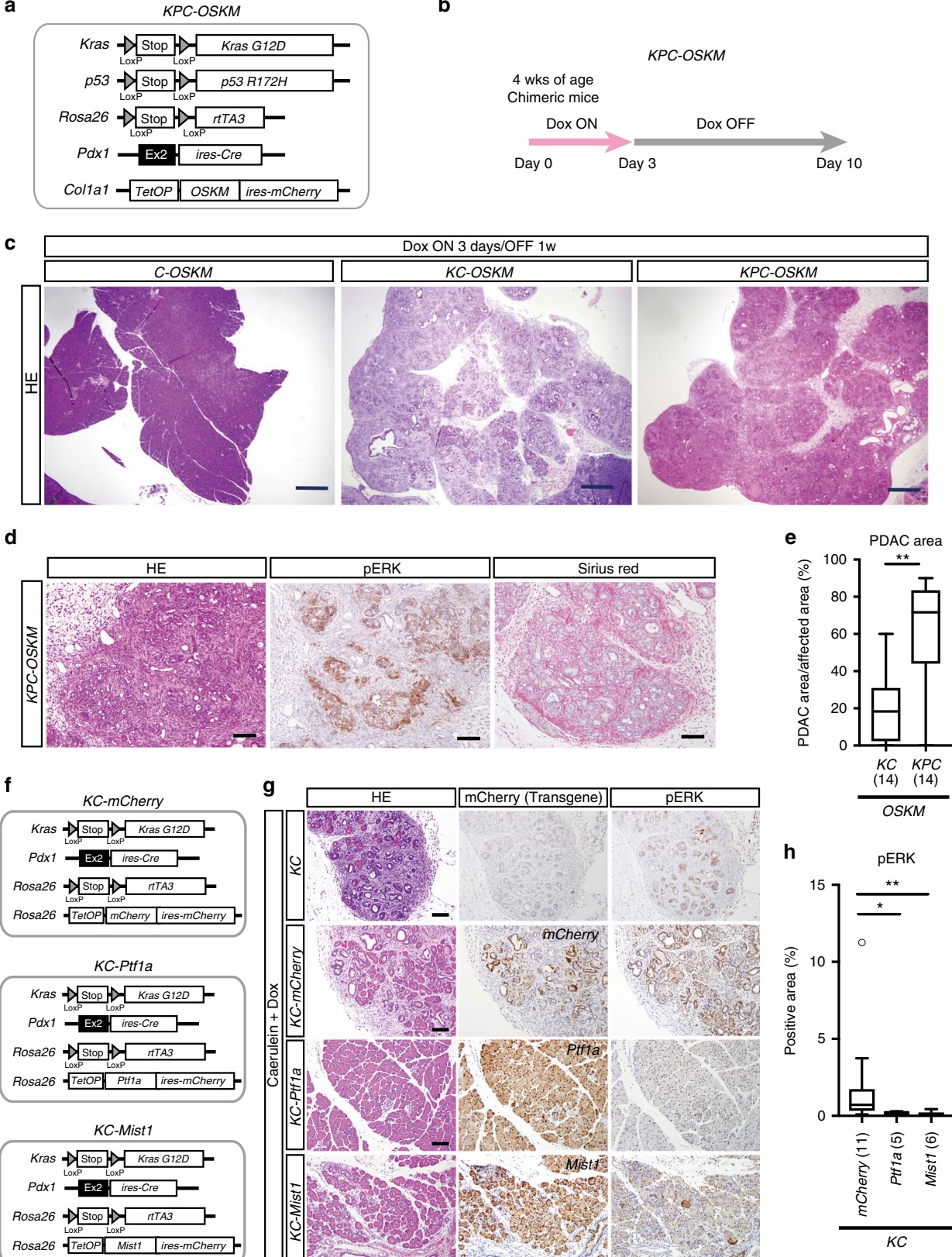

reprogramming in vitro, reprogramming factors repress somatic enhancers, which causes the repression of cell-type specific gene expressions[6]. Consistently, the gradual repression of somatic cell-specific gene expressions is similarly observed during in vivo reprogramming[8]. Notably, the early reprogramming process in vivo is reversible up to 7 days, which reflects epigenetic memory of the cells at early-stage reprogramming[8,46]. In the present study, we show that in vivo reprogramming primarily represses acinar cell enhancers and induces remarkable down-regulation of acinar cell-related genes such as *Ptf1a* and *Mist1*, suggesting that in vivo reprogramming induces the loss of acinar identity. The fact that this loss is reversible further supports the notion that the process is an epigenetic event. Together, the rapid and widespread development of PDAC by 3-day expression of reprogramming factors provide in vivo evidence that transient alteration of epigenetic regulation plays a critical role on cancer initiation. Our results also highlight the significance of dedifferentiation-related epigenetic regulation in cancer initiation, which may provide mechanistic insight into how dedifferentiation is involved in cancer development.

Previous studies using genetically engineered mouse models show that adult ductal cells are refractory to *Kras* mutation-mediated oncogenic transformation. In contrast, adult acinar cells are competent to give rise to PDAC in the presence of *Kras* mutation in which ADM plays a role as an initiating event[20–22]. Although ADM is associated with the initiation of *Kras*-mediated abnormal cell growth, the cause of ADM formation remains to be fully understood. Here we demonstrate that the repression of acinar cell enhancers causes ADM formation, which is consistent with previous findings in which the genetic ablation of acinar cell-related TFs such as *Ptf1a*, *Mist1*, and *Nr5a2* similarly leads to ADM in the pancreas[47–49]. Our results showing that the forced expression of *Ptf1a* or *Mist1* partially prevents reprogramming-induced ADM formation further support the conclusion that the repression of acinar cell enhancers induces ADM formation. Notably, the transient ADM formation becomes irreversible and proceeds to PDAC in *Kras* mutant background, which is accompanied by continuous activation of the ERK signaling pathway. The results suggest that transient repression of acinar cell identity can initiate continuous cancer growth in *Kras*-mutated pancreatic cells. Consistently, forced expression of acinar cell-related TFs (*Ptf1a* or *Mist1*) successfully inhibits *Kras*-mediated activation of the ERK signaling pathway and ADM formation in a caerulein-induced pancreatitis model. Given that ADM is observed in human pancreatitis, a representative risk factor for PDAC[50–52], and pancreatitis promotes PanIN/PDAC formation in mice[53–55], epigenetic fluctuations provoked by inflammatory stimuli may initiate the repression of acinar cell enhancers, which results in continuous cancer cell growth in human pancreas. Collectively, we propose that the stable maintenance of acinar cell enhancers is a roadblock for the cancer cell growth of *Kras*-mutated pancreatic cells.

Gradual activation of the ESC regulatory network occurs during late-stage reprogramming[5,6]. Here we show that prolonged expression of reprogramming factors longer than 1 week results in the development of different types of cancer in the pancreas and stomach that is characterized by the expression of pluripotency-related proteins. Notably, these cancers resemble human AFP-producing cancer, one of the most fatal adult cancers[37,38,56], suggesting that epigenetic reorganization related to late-stage reprogramming may be involved in such cancers. Consistently, human AFP-producing gastric cancers often co-express pluripotency-related proteins, such as SALL4, LIN28A, and LIN28B. Furthermore, these gastric cancers exhibit similar activation patterns of ESC modules as pluripotent stem cells. Our results suggest that the activation of pluripotency-related genes via in vivo reprogramming may be involved in development of certain types of adult cancers.

In summary, taking advantage of in vivo reprogramming systems, we show that reprogramming-related epigenetic regulation has a profound impact on *Kras*-induced cancer. We propose that epigenetic fluctuations provoked by environmental factors may cause a transient dedifferentiation state, which is sufficient for continuous ERK signaling activation and cancer growth of quiescent pancreatic cells with *Kras* mutation.

## Methods

**Vectors.** *Pdx1-ires-Cre* vector: The cDNA fragment (3.8 kbp) of *ires-Cre-pA-PGK-Puro-pA* with 50 bp homologous arms was established using KAPA HiFi Hotstart Readymix (KAPA Biosystems). This fragment was recombined into the 3′ side of the exon 2 of *Pdx1* BAC (BACPAC Resources Center) using the Red/ET BAC recombination system. The fragment of *Pdx1-ires-Cre-pA-PGK-Puro-pA* sequence with 3.8 (5′ arm)/2.5 kb (3′ arm) homologous arm was retrieved and used as a targeting vector. *LSL-HA tag-Kras^G12D* vector: *LoxP-PGK-Bsd-pA-LoxP-HA tag-Kras^G12D* fragment (1.9 kbp) with 350/750 bp homologous arms was generated and recombined into the 5′ side of the untranslated region of the *Kras* BAC (BACPAC Resources Center), retrieved with 4.7 (5′ arm)/4.7 kbp (3′ arm) homologous arm, and used as a targeting vector. *Rosa LSL-rtTA3* vector: rtTA3 cDNA construct with Kozak sequence (753 bp) was generated and inserted into *pROSA26-DEST* vector (Addgene) using Gateway technology (Thermo Fisher Scientific) and used as a targeting vector. pCR8-GW-*SNEL* vector: A 6.4 kbp fragment containing *Sall4-P2A-Nanog-T2A-Esrrb-E2A-Lin28a* cDNA was generated using In-Fusion HD Cloning Kit (Takara Bio). The cloning primers are shown in Supplementary Table 3. The Kozak sequence was added to the 5′ side of *Sall4* cDNA. This fragment was inserted into pCR8-GW-TOPO using Topo cloning technology (Invitrogen) and inserted into *Col1a1* locus of KH2 ESCs using the flp-in recombination system[24]. *Rosa::tetO-mCherry* vector: *mCherry* cDNA (717 bp) was cloned using the primers described in Supplementary Table 3. The Kozak sequence was added to the 5′ side of *mCherry* cDNA. This fragment was inserted into pCR8-GW-TOPO vector using Topo cloning technology (Invitrogen) and inserted into *Rosa::tetO-attR1-ccdB-attR2-ires-mCherry* targeting vector as described previously[57] using Gateway technology. *Rosa::tetO-Mist1* vector: *Mist1* cDNA (600 bp) was cloned using the primers described in Supplementary Table 3. The Kozak sequence was added to the 5′ side of *Mist1* cDNA. This fragment was inserted into pCR8-GW-TOPO using Topo cloning technology (Invitrogen) and

**Fig. 5** *p53* mutation results in rapid development of in vivo reprogramming-induced PDAC in *Kras*-mutated pancreas. **a** A schematic illustration of the genetic construct of *Pdx1 ires-Cre, LSL-Kras^G12D, LSL-p53^R172H, Rosa LSL-rtTA3, Col1a1::tetO-OSKM-ires-mCherry* (KPC-OSKM) mice to induce pancreas-specific *Kras/p53* compound mutations and in vivo reprogramming. **b** Experimental protocol for Dox treatment in KPC-OSKM chimeric mice. **c** Representative images of the pancreas of C-OSKM, KC-OSKM, KPC-OSKM mice given Dox for 3 days, followed by Dox withdrawal for 1 week. Scale bars, 500 μm. **d** pERK and Sirius red staining of the pancreas in KPC-OSKM mice. Scale bars, 100 μm. **e** Quantification of the PDAC area in the pancreas of KC-OSKM and KPC-OSKM chimeric mice. The PDAC area/affected area is shown. A box-and-whisker plot of the PDAC area/affected area. Solid lines in each box indicate the median. Bottom and top of the box are lower and upper quartiles, respectively. Whiskers extend to ±1.5 interquartile range (IQR). Numbers in parentheses indicate the number of mice examined. **p < 0.01, Mann–Whiteney *U*-test. **f** A schematic illustration of the genetic construct of KC mouse with Dox-inducible *mCherry, Ptf1a*, and *Mist1* alleles. **g** *Ptf1a* and *Mist1* induction inhibit caerulein-induced PanIN formation and activation of ERK signaling, but *mCherry* expression does not. Transgenes are induced by Dox treatment starting at 24 h before caerulein treatment. Chimeric mice were sacrificed after 7 days of caerulein treatment. Despite the infiltration of inflammatory cells, PanIN formation and pERK staining are hardly detectable in KC-*Ptf1a* and KC-*Mist1* chimeric mice. Scale bars, 100 μm. **h** Quantification of the pERK-positive area/mCherry-positive area in the pancreas of caerulein-treated KC-*mCherry*, KC-*Ptf1a*, and KC-*Mist1* mice. A box-and-whisker plot of the pERK-positive area/mCherry-positive area. Solid lines in each box indicate the median. Bottom and top of the box are lower and upper quartiles, respectively. Whiskers extend to ±1.5 interquartile range (IQR). Numbers in parentheses indicate the number of mice examined. **p < 0.01, *p < 0.05, Mann–Whiteney *U*-test

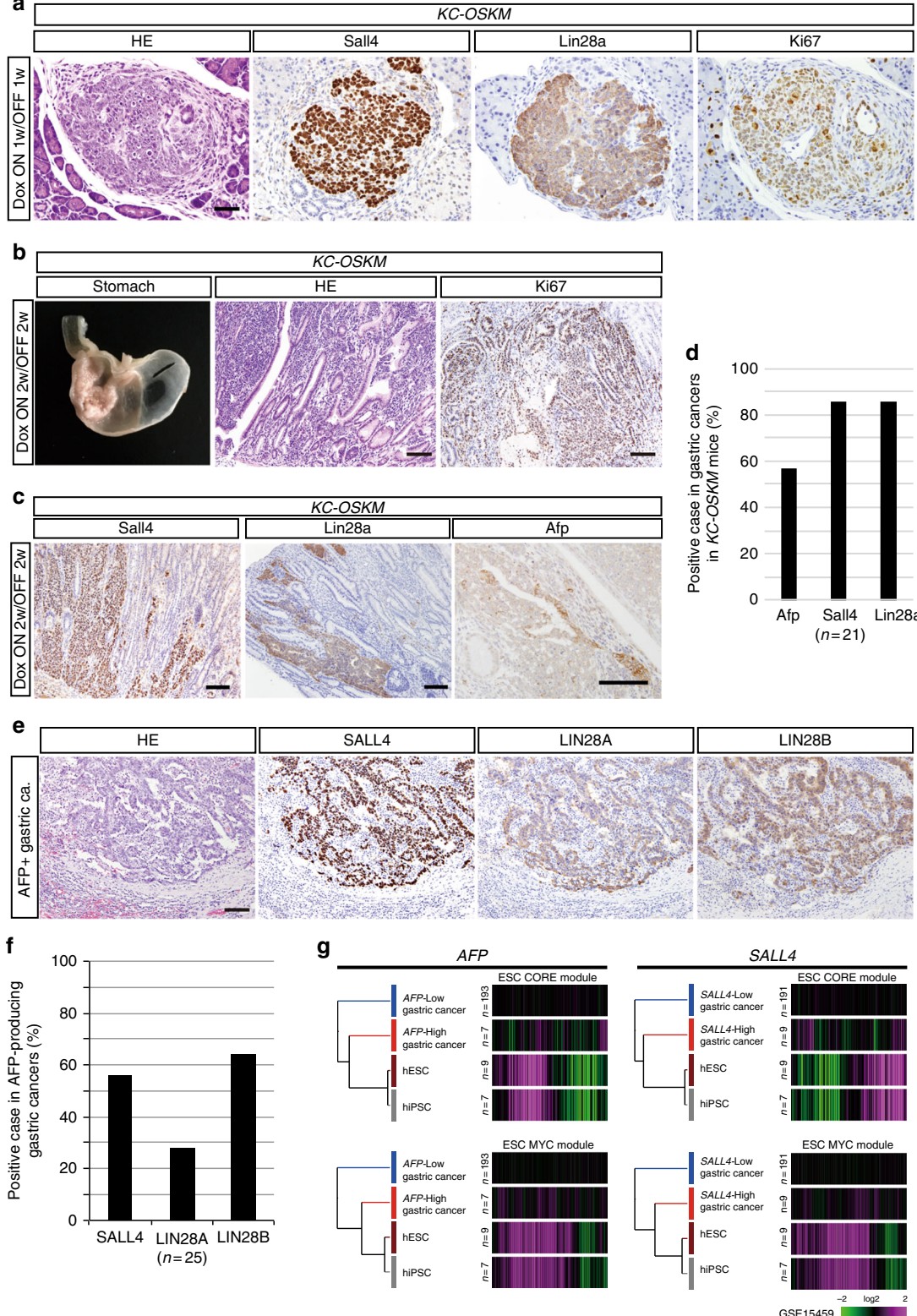

**Fig. 6** Prolonged expression of reprogramming factors induces poorly differentiated cancers in *KC-OSKM* chimeric mice. **a** Sall4, Lin28a, and Ki67 staining in undifferentiated pancreatic cancer induced by Dox treatment for 1 week followed by Dox withdrawal for 1 week in *KC-OSKM* chimeric mice. Scale bar, 50 μm. **b** Representative macroscopic image, histology, and Ki67 immunostaining of gastric cancer induced by prolonged transient expression of reprogramming factors in *KC-OSKM* chimeric mice (Dox On for 2 weeks and then Dox Off for 2 weeks). Scale bars, 100 μm. **c** Immunotaining for Sall4, Lin28a, and Afp in poorly differentiated gastric cancers in *KC-OSKM* chimeric mice. Scale bars, 100 μm. **d** Incidence of Afp, Sall4, and Lin28a expression in poorly differentiated gastric cancers in *KC-OSKM* chimeric mice. **e** Histology and immunostaining for SALL4, LIN28A, and LIN28B in human AFP-producing gastric cancers. Scale bar, 100 μm. **f** Incidence of SALL4-, LIN28A-, and LIN28B-positive cases in human AFP-producing gastric cancers. **g** Partial activation of ESC-Core and ESC-Myc modules in human gastric cancers expressing *AFP* or *SALL4*

inserted into *Rosa::tetO-attR1-ccdB-attR2-ires-mCherry* targeting vector by Gateway technology. *Rosa::tetO-Ptf1a* vector: *Ptf1a* cDNA was synthesized using a commercial service (Eurofines genomics). The Kozak sequence was added to the 5′ side of *Ptf1a* cDNA. The cDNA fragment which was cut by *Eco*RI and purified with Gel/PCR Extraction Kit (Fast Gene), inserted into pCR8-GW-TOPO vector, and recombinated into *Rosa::tetO-attR1-ccdB-attR2-ires-mCherry* targeting vector by Gateway technology. *PiggyBac tetO-EGFP-ires-EGFP* vector: The *EGFP* cDNA (720 bp) and kozac sequence was cloned and inserted into pCR8-GW-TOPO using Topo cloning technology (Invitrogen) and inserted into *PiggyBac tetO- attR1-ccdB-attR2-ires-EGFP* vector (KW509) using Gateway technology. *PiggyBac tetO-Ptf1a-ires-EGFP* vector: The *Ptf1a* cDNA from *pCR8-Ptf1a-GW* vector described above was inserted into *PiggyBac tetO- attR1-ccdB-attR2-ires-EGFP* vector (KW509) using Gateway technology. *PiggyBac tetO-Mist1-ires-EGFP* vector: The *Mist1* cDNA from *pCR8-Ptf1a-GW* vector described above was inserted into *PiggyBac tetO- attR1-ccdB-attR2-ires-EGFP* vector (KW509) using Gateway technology.

**Homologous recombination into ES cells**. The ESC lines V6.5 and KH2 were previously described .[24,58] Twenty micrograms of each targeting vector were linearized using specific restriction enzymes (*Pdx1-ires-Cre* targeting vector: *Sca*I, *LSL-HA tag-Kras*$^{G12D}$ targeting vector: *Asi*SI, *Rosa LSL-rtTA3* targeting vector: *Pvu*I, *Rosa::tetO-mCherry* targeting vector: *Asi*SI, *Rosa::tetO-Mist1* targeting vector: *Asi*SI, *Rosa::tetO-Ptf1a* targeting vector: *Asi*SI). All restriction enzymes were obtained from New England Biolabs. After 37 °C overnight linearization, each product was purified and collected by ethanol precipitation. The pellets were dissolved into 100 μL of 25 mM HEPES buffer (Gibco). ESCs were dissociated with 0.25% trypsin (Nacalai tesque) and dissolved into 500 μL of high glucose DMEM (Nacalai tesque) containing 25 mM HEPES. A mixture of the targeting vector and ESCs ($4.0 \times 10^6$) were injected into Gene Pulser Cuvette (BIO-RAD) and electroporation was performed with Gene Pulser Xcell (BIO-RAD). The electroporation was set at 550 V, 600 ms and performed twice. After electroporation, $2.0 \times 10^6$ ESCs were placed over MEFs in a 6 cm dish (total 2 dishes) and cultured at 37 °C with 5% $CO_2$ in ESC medium (Knockout DMEM (Gibco), 2 mM L-glutamine (Nacalai tesque), 1× NEAA (Nacalai tesque), 100 U/mL penicillin, and 100 μg/mL streptomycin (Nacalai tesque), 15% fetal bovine serum (Gibco), 0.11 mM mercaptoethanol (Gibco), and 1000 U/mL human LIF (Wako)). After 24 h of electroporation, we started the selection with the specific antibiotics. The concentrations of antibiotics used in the present study were blasticidinS (Funakoshi) 15 μg/mL, puromycin (Sigma) 1 μg/mL, G418 (Nacalai tesque) 350 μg/mL, and hygromycin B (Roche) 150 μg/mL. After 1 week of antibiotics selection, the survived ESC colonies were picked up and expanded to establish ESC lines.

**Southern blotting**. Genomic DNA was collected from ESCs or pancreatic tissues using the Purelink Genomic DNA mini Kit (Invitrogen). The recombinated BAC plasmids were collected using NucleoBond Xtra BAC Kit (Marchery-Nagel). The extraction was performed according to the manufacturer's instructions. Three nanograms of each recombinated BAC as a positive control and 5 μg of sample genomic DNA were digested with specific restriction enzymes (*Pdx1-ires-Cre*: *Sca*I, *LSL-HA Kras*$^{G12D}$: *Nhe*I, *LSL-Kras*$^{G12D}$: *Mfe*I, *LSL-p53*$^{R172H}$: *Nhe*I) at 37 °C overnight. The restriction enzymes were obtained from New England Biolabs. The restricted products were ethanol precipitated and then dissolved into 15 μL of TE buffer. The products were mixed with 5 mL of 6× Loading Buffer Triple Dye (NIPPON Gene), and electrophoresis was performed in 0.8% agarose gel (Fast Gene), 50 V for 2 h. The products in the gel were depurinated in 250 mM HCl for 10 min, washed in distilled water for 5 min, and denaturated in 0.5 M NaOH, 1.5 M NaCl for 15 min two times. The products were then transferred to Hybridization Transfer Membrane (PerkinElmer) by capillary transfer at room temperature for 8 h. The membrane was washed with 2× Standard Saline Citrate buffer (SSC buffer) for 5 min twice and crosslinked with CL-1000 Ultraviolet Crosslinker (UVP). The samples were fixed at pH 7.0 using HCl. The membrane was hybridized at 68 °C overnight with 25 ng/mL probes in PerfectHyb buffer (TOYOBO). Probes were generated using the PCR DIG Probe Synthesis Kit (Roche) with wild-type BAC as the template. The primer sets of each target gene are shown in Supplementary Table 4. The membrane was washed with low stringency buffer (containing 2× SSC buffer and 1× SDS) for 5 min twice and high stringency buffer (containing 0.2× SSC buffer, 1× SDS) for 5 min twice. Next, DIG Luminescent Detection Kit for Nucleic Acids (Roche) was used for visualization, and LAS4000 (GE Healthcare) was used for detection.

**Blastocyst collection and microinjection**. Blastocysts collection: 8-week-old ICR female mice (Japan SLC) were treated with 7.5 U of serotropin (ASKA Animal Health) by intraperitoneal injection. After 48 h of serotropin treatment, mice were injected 7.5 U of gonatropin (ASKA Pharmaceutical). These mice were then mated with ICR male mice (Japan SLC). Plug check was performed the next morning. Two days later, these female mice were sacrificed by cervical dislocation and operated to pick out oviducts. Two-cell stage fertilized eggs were collected by perfusion with M2 medium (Sigma) and maintained in KSOM medium. Two days later, the obtained blastocysts were used for microinjection. For microinjection, ESCs were treated with trypsin and pipetted 15 times to dissociate them into single cells. The single cells were cultured in a gelatin-coated 10-cm dish with 10 mL ESC medium for 30 min to attach MEFs onto the dish. Three milliliters of supernatant containing ESCs was collected. Three to five ESCs were injected into the ICR blastocysts under an OLYMPUS IX71 microscope. Twenty to 25 injected blastocysts were transplanted into the uterus of pseudopregnant ICR female mice (Japan SLC).

**Establishment of ESCs**. Collected blastocysts were placed on the MEFs in a 24-well dish and cultured with 500 μL of 2i-containing ESC medium (1 μM of PD0325901 (STEMGENT) and 3 μM of CHIR99021 (STEMGENT) in ESC medium)[59]. One hundred microliters of 2i-ESC medium was added into each well every day. One week later, the colonies were passaged and expanded in ESC medium on the MEFs. Male ESCs were selected by genotyping *Sry* allele using Go taq Master Mix (Promega). Genotyping primers are described in Supplementary Table 5.

**Mice**. *Rosa LSL-LacZ* mice[16], *LSL-Kras*$^{G12D}$ mice[18], *LSL-p53*$^{R172H}$ mice[19], and *Col1a1::tetO-OSKM-ires-mCherry* mice[8,60] have been described previously. All mice were maintained on a C57BL/6 and 129X1/Sv mixed background. For germline transmission, 8-week-old male chimeric mice were mated with 8-week-old C57BL/6 female mice (Japan SLC) to obtain transgenic mice. All chimeric mice were generated by injecting male ES cells which have C57BL/6 and 129X1/Sv mixed genetic background into blastocysts of ICR mice.

**Mice experiment**. All animal studies were conducted in compliance with ethical regulations in Kyoto University and were approved by Center for iPS Cell Research and Application (CiRA), Kyoto University. To obtain the unbiased and reliable results, more than five mice were used in each group. Sample size was indicated in each figure. A few mice had been excluded because of the mortality before the evaluation. The coat color chimerism of chimeric mice was evaluated before experiments (Supplementary Fig. 5B). Mice that had no chimeric contribution had been excluded in each experiment.

**Mice genotyping**. Ear tips of 3-week-old mice were collected and dissolved in 500 μL of DNA elution buffer (containing 100 mM Tris HCl, 5 mM EDTA, 0.2% SDS, 200 mM NaCl, and 1% Protein kinase) at 65 °C. After centrifugation at 4 °C and 15,000 rpm for 5 min, 300 μL of supernatant was collected and mixed with equivalent isopropyl alcohol and vortexed. After centrifugation at 4 °C and 15,000 rpm for 10 min, the pellet was mixed with 70% of ethanol and precipitated. Then the pellet was dissolved in 50 μL of Tris-EDTA buffer. Recombination was detected by PCR with Go taq Master Mix and Polymerase (Promega). Genotyping Primers are shown in Supplementary Table 5.

**Alkaline phosphatase staining of Dox-treated KH2-*SNEL* MEFs**. MEFs from E13.5 of KH2-*SNEL* chimeric mice were cultured with 2.0 μg/mL of Dox in ESC medium for 2 weeks. ALP staining was performed according to the manufacturer's instructions using the ALP stain Kit (Muto Pure Chemicals).

**X-gal staining**. The precise procedure was described previously[61]. Whole mount staining: Mice under anesthesia were perfused through the vascular system with 10 mL of phosphate-buffered saline (PBS) to clear the blood. Perfusion was performed with 10 mL of 4% PFA to obtain adequate fixation. The dissected pancreas of *Pdx1-ires-Cre, Rosa LSL-LacZ* mice were collected and fixed with ice-cold 4% PFA for 1 h. The tissues were poured into PBS containing 10% of sucrose, and then into PBS containing 20% of sucrose (Nacalai Tesque) for 1 h, respectively, and then placed in PBS containing 30% of sucrose at 4 °C for overnight. Next the pancreas was poured into permeabilization solution (5 mM EGTA, 2 mM $MgCl_2$, 0.01% sodium deoxycholate, 0.02% Nonidet P-40 in PBS) and reacted with the X-gal solution (1 mg/mL X-gal, 5 mM potassium ferrocyanide, and 5 mM potassium ferricyanide in the permeabilization solution) at 37 °C for 6 h. Frozen tissue staining: After fixation of the dissected tissues with PBS containing 30% sucrose, the pancreas was embedded in Tissue-Tek O.C.T compound (Sakura). Frozen tissues were cut serially into 10-μm-thick sections and stained with the procedure described above. Counterstaining was performed using Nuclear Fast Red (Merck Millipore).

**Doxycycline treatment**. Doxycycline (Sigma) was dissolved in drinking water at 2 mg/mL. For quantitative PCR (qPCR) and ChIP-seq of the pancreas of *C-OSKM* mice, Dox-containing PBS was intraperitoneally injected to induce uniform transgene expressions (1 mg/injection every consecutive 12 h until sacrifice). To maintain acinar cell enhancer in *Ptf1a* or *Mist1*-inducible mice, we started Dox administration to induce *Ptf1a* or *Mist1* 24 h before caerulein treatment.

**Caerulein treatment**. The precise procedure was described previously[28]. We performed intraperitoneal injection of caerulein (Sigma) eight times for 2 days (2 μg/injection every consecutive hour for 7 h). Caerulein was dissolved in PBS.

**RNA extraction from tissues**. The tissues of mice were freshly collected and dipped into 1 mL of RNA later (Invitrogen) overnight at 4 °C. Subsequently, the tissues were poured into 1 mL of Sepazol (Nacalai tesque) and shattered with a sonicator (Qsonica) for 5 s. After centrifugation at 4 °C and 12,000 $g$ for 10 min, supernatants were carefully collected and mixed with 200 μL of chloroform (Wako). After shaking for 15 s, the mixtures were left to stand for 2 min at room temperature and centrifuged at 4 °C and 12,000 $g$ for 15 min. Three hundred microliters of supernatant was carefully collected and mixed with 300 μL of 70% ethanol. The mixtures were injected into RNeasy spin column (Qiagen), and RNA was extracted according to the manufacturer's instructions.

**Quantitative RT-PCR**. RNA was extracted using RNeasy Plus Mini Kit (Qiagen) according to the manufacturer's protocol. RNA was quantified with NanoDrop (Thermo Scientific). Five-hundred nanograms of RNA was used for the reverse transcription reaction into cDNA by using PrimeScript RT reagent Kit (Takara Bio). Real-time qPCR was performed by using Go Taq qPCR Master Mix (Promega). Transcript levels were normalized by $\beta$-actin. The PCR primers are shown in Supplementary Table 6.

**Western blotting from the pancreatic tissue**. The pancreatic tissues were sectioned into 5 × 5 mm and shattered with a sonicator (Qsonica) for 5 s in RIPA buffer (10 mM of Tris HCl (pH 8.0), 150 mM HCl, 1% Triton, 1% DOC, and 0.1% SDS) containing 0.5% Protease inhibitor, 1% DTT and 100× phosphatase inhibitor (Nacalai tesque) on ice. After centrifugation at 20,000 $g$ for 10 min, the 100 μL of supernatants were collected. The quantity was determined with a Molecular Devices VersaMax Microplate Reader (Marshall Scientific) using Quick start protein assay Kit (BIO-RAD). The samples were processed in 30 μg, and denatured at 95 °C for 5 min. Electrophoresis was performed in 10% SDS-PAGE gel, 100 for 15 and 200 V for 35 min with the use of PowerPac HC (BIO-RAD). Next, the protein in the gel was transferred onto the Amersham Hybond-P PVDF Membrane (GE HealthCare). Membranes were blocked for 60 min with 4% Skim Milk (Nacalai Tesque) in 1× TBS with 0.05% Tween-20 (TBST) and then incubated with primary antibodies in blocking buffer (4% Skim Milk in TBST) overnight at 4 °C. The lists of antibodies and dilutions are shown in Supplementary Table 7. The membrane was washed in TBST and incubated with secondary antibodies in blocking buffer for 1 h at room temperature. The membrane was washed in TBST again, then incubated with Pierce ECL plus Western Blotting Substrate (Thermo Scientific) for visualization. The bands were detected with LAS4000 (GE HealthCare).

**Paraffin-embedded specimen preparation**. The dissected tissue samples were fixed in 4% PFA (Wako) for overnight at room temperature. The next day, the samples were poured into 70% ethanol (Wako) for several hours, and then poured into 80% ethanol, 90% ethanol, 100% ethanol, and xylene (Wako), and embedded in paraffin using a spin tissue processor (STP120; Thermo Scientific). Sections were sliced into 3–4 μm thickness, and stained with hematoxylin and eosin (Muto Pure Chemicals). Serial sections were used for the immunohistochemical analyses.

**Immunostaining**. The samples were soaked in xylene for 5 min three times to remove paraffin and 100% ethanol for 5 min three times to hydrophilize. After washing with water for several min, the samples were soaked in epitope retrieval buffer (Nichirei Bioscience) and microwaved at 100 W for 10 min. The samples were then soaked in 1× PBS for several min, and incubated with 150 μL primary antibodies in PBS with 2% bovine serum albumin (BSA) (MP Biomedicals) for 1 h at room temperature. The lists of antibodies and dilutions are shown in Supplementary Table 7. After washing with PBS for 5 min two times, the samples were processed with secondary antibodies for 30 min at room temperature. The samples were washed in PBS for 5 min two times, and nuclear staining was performed with hematoxylin for 10–30 s. After washing with water for 10 min, the samples were poured into 100% ethanol three times and placed in xylene three times. The samples were evaluated under a microscope (Olympus BX41).

**Immunofluorescent staining**. The same procedure as immunostaining described above was performed until primary antibodies treatment. The samples were processed with 500× DAPI (Invitrogen) and 150× fluorescence-labeled secondary antibodies diluted with 0.5% BSA PBS, for 90 min at room temperature. The lists of antibodies and dilutions are shown in Supplementary Table 7. After washing in PBS for 5 min two times, the samples were mounted and evaluated with a confocal laser scanning microscope (Zeiss LSM700).

**Alcian blue staining**. Tissue samples were soaked into xylene for 5 min three times, 100% ethanol for 5 min three times, washed with water for several minutes, and finally stained with Alcian blue 8GX solution (Merck Millipore) and nuclear fast red (Merck Millipore), according to the manufacturer's protocol.

**Sirius red staining**. Tissue samples were soaked in xylene for 5 min three times and 100% ethanol for 5 min three times, washed with water for several minutes, and stained with Picro-Sirius Red Stain Kit (PSI).

**Microarray analysis**. Two hundred nanograms of total RNA prepared from the pancreatic tissues was subjected to cDNA synthesis with a WT Expression Kit (Ambion), and the resultant cDNA was fragmented and hybridized to a Mouse Gene 1.0 ST Array (Affymetrix). After hybridization, GeneChip arrays were washed and stained using GeneChip Fluidics Station 450 (Affymetrix) and detected with the Scanner 3000 TG system (Affymetrix) following the manufacturer's standard protocols. The data were analyzed using GeneSpring GX software (version 14.8; Agilent Technologies).

**Chromatin immunoprecipitation from pancreatic tissue**. Perfusion fixation with ice-cold 4% PFA (Wako) was employed for fixation of the pancreatic tissue of C-OSKM mice. The pancreas was dissected after 5 min of fixation and treated with glycine to quench the formaldehyde. ChIP was performed as described previously[62]. H3K27Ac antibody (Monoclonal Antibody Lab) was used for the ChIP analysis, and mouse IgG antibody (Abcam) was used as a control.

**ChIP-seq**. Purified ChIP DNA (5 ng per sample) was used for the generation of sequencing libraries with TruSeq ChIP Sample Prep Kit (Illumina). The resultant DNA libraries were assessed on Agilent Bioanalyzer and quantified with KAPA Library Quantification Kits (KAPA BIOSYSTEMS). The libraries were sequenced to generate single-end 86 bp reads using NextSeq 500 (illumina).

**ChIP-seq data analysis**. ChIP-Seq Peak calls were made as previously described[63]. Briefly, the low-quality bases at the 3′ read ends and the adaptors were trimmed using cutadapt-1.9.1 (ref. [63]). Untrimmed and trimmed reads of 20 bp or longer were mapped to mouse genome (mm10) with BWA 0.7.12 (ref. [64]). Duplicate reads and low mapping quality reads (MAPQ < 20) were removed using Picard (http://broadinstitute.github.io/picard/) and SAMtools[65] for further analysis. Mapped reads were visualized using igv-2.3 (ref. [66]). Peaks were called using MACS2 (ref. [67]) plus IDR framework developed by the ENCODE project[68]. The reads mapped to the blacklisted regions (https://sites.google.com/site/anshulkundaje/projects/blacklists) and sex chromosomes were excluded for the peak calling. The heatmaps for ChIP-Seq data were drawn using the ngsplot-2.47 program[69]. Super enhancers were identified by H3K27ac with ROSE pipelines[29,30]. Super enhancers identified in this study were assigned to the transcripts whose TSS were within the SEs.

**Bisulfite sequencing from pancreatic tissue**. Freshly collected pancreatic tissues were frozen in liquid nitrogen and grounded into powder using mortar. Genome DNA of the pancreas was extracted using Purelink Kit (Invitrogen) according to the manufacturer's protocol. Bisulfite treatment was performed using EZ DNA Methylation-Gold Kit (Zymo Research). The target sequence was cloned by Ex Taq HS (Takara Bio) and ligated into pCR4-Topo vector using Topo cloning technology (Invitrogen). These plasmids were transformed into DH5α at 37 °C, overnight. Colony PCR was performed using Go taq Master Mix (Promega) and ascertained ligation of the cassettes. After cleanup of the PCR products by exonuclease I (New England Biolabs) and shrimp alkaline phosphatase (Takara Bio), the sequence reaction was performed using M13 Rv primer (5′-CAGGAAA-CAGCTATGAC-3′) and was sequenced with ABI 3500 xL (Applied Biosystems). Methylation rates were measured using Quantification tool for Methylation Analysis (QUMA) software (Riken).

**Quantification analysis**. To quantify the immunostained tissue samples, immunostained section was randomly photographed at ×200 magnification three times for each mouse. The pictures were processed with ImageJ software (NIH) to evaluate the positive area of each CK19, phospho-ERK, Alcian blue, Sirius red, and EGFR staining. The positive area was determined as the mean of positive area in three independent histological images. The results were evaluated with Graphpad Prism 7 software for quantification. To assess γH2AX-, p21-, Ki67-, Caspase 3-positive rates, we randomly photographed epithelial area of the pancreas at ×200 magnification. The positive nucleus of epithelial cells was counted. The number of each positive nucleus was divided by the number of total nucleus of epithelial cells. For quantification of immunofluorescent samples, section was randomly photographed at ×200 magnification. The ratio of CK19-positive cells in EGFP-expressing cells was determined in ten independent immunofluorescent images for each cohort.

**Statistical analysis of tissue samples**. Statistical parameters including statistical significance and $n$ values are described in the figures. Statistical analyses were carried out using Prism 7 Software (GraphPad). For statistical comparison of two groups, we performed Mann–Whiteney $U$-test (two-tailed, 95% confidence intervals). A box-and-whisker plot is used to display information about the range, the median and the quartiles. Solid lines in each box indicate the median. Bottom and top of the box are lower and upper quartiles, respectively. Whiskers extend to ±1.5 interquartile range (IQR). Numbers in parentheses indicate the number of mice examined. *$p < 0.05$, **$p < 0.01$, ***$p < 0.001$. n.s., not significant.

**Patients-derived paraffin-embedded tissue samples.** Human patients-derived paraffin-embedded tissue samples were used in compliance with ethical regulations in Kyoto University Hospital.

**Statistical analysis of survival.** Kaplan–Meier survival curves were calculated using the survival time for each mouse. The Log-rank test was used to test for significant differences between the groups using Prism 7 Software (GraphPad).

**Data availability.** ChIP-seq and microarray data reported in this study have been deposited to the Gene Expression Omnibus (GEO) under accession GSE100842.

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

## Acknowledgements

We are grateful to P. Karagiannis for critical reading of this manuscript. The authors were supported in part by Cancer research grant P-CREATE, Japan Agency for Medical Research and Development (AMED) (JP18cm0106203); AMED-CREST, AMED (JP18gm1110004); SICORP, AMED (JP17jm0210039); Core Center for iPS Cell Research, Research Center Network for Realization of Regenerative Medicine, AMED; JSPS KAKENHI Grant Number 15H04721; the Princess Takamatsu Cancer Research Fund; the Takeda Science Foundation; and the Naito Foundation.

## Author contributions

H.S., N.S., A.T., T.U., M.K., S.S., B.K., K.W., Y.I., Y.K., T.Y. and Yasuhiro Y. conceived the ideas and designed the experiments, acquired the data and performed the analyses and interpretations. S.K., Yosuke Y., N.S., A.T., T.U., T.Y., and Yasuhiro Y. provided technical assistance. S.K. and A.T. assisted with mouse experiments. Yosuke Y. and H.H. provided clinical samples and contributed to clinical data analyses. N.S., K.W., and Y.K. provided materials. N.S., Y.Y., H.H., and Yosuke Y. contributed to pathological examination. H.S. and Yasuhiro Y. wrote the manuscript.

## Additional information

**Competing Interests:** The authors declare no competing interests.

