## [Peer Review File · Nature Communications]

Reviewers' comments:

Reviewer #1 (Remarks to the Author):

As I noted on my earlier review, the premise of the paper is based on some wonderful work by these authors. It didn't have quite the impact of the other because of the Wei paper on KLF4. The authors address this argument very well that they are looking at a reversible ADM phenotype, and the epigenetic dynamics. I also noted that there was a gene tour de force nature of this paper, and they have now quantified pERK-positive cells in caerulein-treated Mist1-, Ptf1a-, and mCherry-inducing KC pancreas. They conclude that maintenance of the acinar cell enhancer can prevent initiation of K-ras induced cancer, which is exciting. They support this conclusion with additional ChIP-seq data on caerulein-treated pancreas. They have also substantially clarified their discussion, which had quite frankly confused me in the earlier version. I think they make a valid argument in response to Reviewer 2's criticism of the TF's they use, the point being that they are suppressing somatic enhancers. It's a similar criticism from Reviewer 3. That reviewer was also impressed though with the amount of work. In summary, this is a very complex area but there is no question that there are important new insights, these are extremely valuable models, and it is well worth sharing with the community.

Reviewer #3 (Remarks to the Author):

In my initial review, I indicated that the prior work in the field relating to ADM and the unclear physiological relevance of the model system limited the overall significance of the manuscript. Nevertheless, the studies are of good quality and are potentially suitable for a journal such as Nature Communications.

Other:

Is the SNEL ADM phenotype reversible? (suggested in lines 182-184 but not shown)

Please clarify the meaning of this (lines 249-250): "Furthermore, ductal lesions were histologically displayed PanIN/PDAC (Fig. 4D, E)." The images seem to be of PanIN-like lesions. Was there clear evidence of higher grade dysplasia?

Are the figures mixed up? The reference to the Supplementary Figure 4E, F on line 253 appears incorrect.

The effects of forced expression of Mist1 and Ptf1a on OSKM-induced ADM are not quantified and no control samples are shown in the figures for comparison.

RE: NCOMMS-18-01846-T

"In vivo reprogramming drives Kras-induced cancer development"

Response to referee's comments:

We thank the reviewers for their helpful comments and suggestions. We have revised our manuscript to address their questions. We have responded to each reviewer's point individually in the subsequent section. We hope that our responses will clarify any remaining concerns and that our manuscript is now suitable for publication.

Reviewer #1 (Remarks to the Author):

As I noted on my earlier review, the premise of the paper is based on some wonderful work by these authors. It didn't have quite the impact of the other because of the Wei paper on KLF4. The authors address this argument very well that they are looking at a reversible ADM phenotype, and the epigenetic dynamics. I also noted that there was a gene tour de force nature of this paper, and they have now quantified pERK-positive cells in caerulein-treated Mist1-, Ptf1a-, and mCherry-inducing KC pancreas. They conclude that maintenance of the acinar cell enhancer can prevent initiation of K-ras induced cancer, which is exciting. They support this conclusion with additional ChIP-seq data on caerulein-treated pancreas. They have also substantially clarified their discussion, which had quite frankly confused me in the earlier version. I think they make a valid argument in response to Reviewer 2's criticism of the TF's they use, the point being that they are suppressing somatic enhancers. It's a similar criticism from Reviewer 3. That reviewer was also impressed though with the amount of work. In summary, this is a very complex area but there is no question that there are important new insights, these are extremely valuable models, and it is well worth sharing with the community.

We thank this reviewer for his/her enthusiasm.

Reviewer #3 (Remarks to the Author):

In my initial review, I indicated that the prior work in the field relating to ADM and

the unclear physiological relevance of the model system limited the overall significance of the manuscript. Nevertheless, the studies are of good quality and are potentially suitable for a journal such as Nature Communications.

We thank this reviewer for his/her enthusiasm.

Other:

Is the SNEL ADM phenotype reversible? (suggested in lines 182-184 but not shown)

In the revised manuscript, we have added the histological image demonstrating the reversible ADM phenotype by the transient *SNEL* expression (Supplementary Figure 2G).

Please clarify the meaning of this (lines 249-250): “Furthermore, ductal lesions were histologically displayed PanIN/PDAC (Fig. 4D, E).” The images seem to be of PanIN-like lesions. Was there clear evidence of higher grade dysplasia?

***KC-OSKM* mice develop both PanIN-like lesions and PDAC lesions. We have added the histological image that indicate higher grade dysplasia in *KC-OSKM* mice (Supplementary Figure 5A).**

Are the figures mixed up? The reference to the Supplementary Figure 4E, F on line 253 appears incorrect.

We have corrected the reference to the Supplementary Figures 4E and F.

The effects of forced expression of *Mist1* and *Ptf1a* on OSKM-induced ADM are not quantified and no control samples are shown in the figures for comparison.

We have quantified CK19-positive cells after forced expression of *EGFP* (control), *Mist1*, and *Ptf1* in Figure 3F.

REVIEWERS' COMMENTS:

Reviewer #3 (Remarks to the Author):

this revised study is suitable for publication in Nature Communications